# Robotic Stereotactic Radiotherapy for Intracranial Meningiomas—An Opportunity for Radiation Dose De-Escalation

**DOI:** 10.3390/cancers15225436

**Published:** 2023-11-16

**Authors:** Hanna Grzbiela, Elzbieta Nowicka, Marzena Gawkowska, Dorota Tarnawska, Rafal Tarnawski

**Affiliations:** 1III Radiotherapy and Chemotherapy Clinic, Maria Sklodowska-Curie National Research Institute of Oncology, Gliwice Branch, Wybrzeze Armii Krajowej 15, 44-100 Gliwice, Poland; 2Institute of Biomedical Engineering, Faculty of Science and Technology, University of Silesia in Katowice, 75 Pulku Piechoty 1A, 41-500 Chorzow, Poland

**Keywords:** meningiomas, robotic stereotactic radiotherapy, CyberKnife, dose de-escalation

## Abstract

**Simple Summary:**

Meningiomas are among the most common tumors that develop inside the skull. They are often treated with radiotherapy, but there is still no agreement on optimal radiation dose. The aim of our study was to assess the effects of CyberKnife radiotherapy with the total dose of 18 Gy delivered in three fractions. We achieved local control in 91.7% of patients and the results were similar to radiotherapy schemes with greater biologically effective dose, which supports the idea of dose de-escalation in the treatment of meningiomas.

**Abstract:**

Objective: To evaluate the possibility of dose de-escalation, with consideration of the efficacy and safety of robotic stereotactic CyberKnife radiotherapy in patients diagnosed with intracranial meningiomas. Methods: The study group consisted of 172 patients (42 men and 130 women) treated in III Radiotherapy and Chemotherapy Clinic of Maria Sklodowska-Curie National Research Institute of Oncology in Gliwice between January 2011 and July 2018. The qualification for dose de-escalation was based on MRI (magnetic resonance imaging) features: largest tumor diameter less than 5 cm, well-defined tumor margins, no edema, and no brain infiltration. The age of patients was 21–79 years (median 59 years) at diagnosis and 24–80 years (median 62 years) at radiotherapy. Sixty-seven patients (Group A) were irradiated after initial surgery. Histopathological findings were meningioma grade WHO 1 in 51 and WHO 2 in 16 cases. Group B (105 patients) had no prior surgery and the diagnosis was based on the typical features of meningioma on MRI. All patients qualified for the robotic stereotactic CyberKnife radiotherapy, and the total dose received was 18 Gy in three fractions to reference isodose 78–92%. Results: Follow-up period was 18 to 124 months (median 67.5 months). Five- and eight-year progression free survival was 90.3% and 89.4%, respectively. Two patients died during the follow-up period. Progression of tumor after radiotherapy was registered in 16 cases. Four patients required surgery due to progressive disease, and three of them were progression free during further follow-up. Twelve patients received a second course of robotic radiotherapy, 11 of them had stable disease, and one patient showed further tumor growth but died of heart failure. Crude progression free survival after both primary and secondary treatment was 98.8%. Radiotherapy was well-tolerated: acute toxicity grade 1/2 (EORTC-RTOG scale) was seen in 10.5% of patients. We did not observe any late effects of radiotherapy. Conclusion: Stereotactic CyberKnife radiotherapy with total dose of 18 Gy delivered in three fractions showed comparable efficacy to treatment schedules with higher doses. This could support the idea of dose de-escalation in the treatment of intracranial meningiomas.

## 1. Introduction

Up to 30% of all primary intracranial tumors are meningiomas, which makes them the most common non-glial tumors in this location [1,2,3]. It is difficult to estimate the real prevalence of meningiomas, as many of them are asymptomatic and are found accidentally or during an autopsy, so the incidence is probably much higher than shown in most registries. Meningiomas are mostly benign tumors. They progress very slowly, and only some of them can affect the patient’s quality of life. As stated in National Comprehensive Cancer Network (NCCN) Guidelines version 2.2022, small and asymptomatic tumors may be observed using serial imaging [4]. Standard therapy in other cases is usually surgical excision, but not all tumors can be completely removed. Some will recur even after GTR (gross total resection), and for some patients, neurosurgery is impossible (due to tumor location or the patient’s poor performance status) [5,6]. For these patients, radiation therapy is a valuable treatment alternative.

Decisions regarding whether the patient requires treatment or can be observed should be made during multidisciplinary board meetings. Factors influencing these decisions are patient-related (age, performance score, comorbidities, personal treatment preferences) or tumor-related (diameter, WHO grade, growth rate, location, proximity to critical structures, presence and severity of symptoms and potential for causing neurological deficits if untreated). Other factors which should be considered are possible neurological complications following surgery or radiotherapy, likelihood of complete resection and/or complete irradiation with stereotactic radiosurgery, further treatment options if progression occurs, and available surgical or radiation oncology expertise and resources.

Radiation therapy is a useful treatment option for benign meningiomas. It should be stressed, however, that in the case of these benign tumors we do not expect a complete response. The main objective of radiation therapy is to stop tumor progression; in some cases, partial response can be achieved. According to NCCN Guidelines, optimal dosing has not been determined [4]. The idea of the current study was thus to set up an observational study using the lowest commonly accepted radiation dose—18 Gy—in three fractions to reference isodose 78–92%.

## 2. Materials and Methods

### 2.1. Patient Characteristics

The study group consisted of 172 patients (42 men and 130 women) diagnosed with intracranial meningiomas, treated in the III Radiotherapy and Chemotherapy Clinic of Maria Sklodowska-Curie National Research Institute of Oncology in Gliwice. Between January 2011 and July 2018 all patients received 18 Gy in three fractions (to isodose line 78–92%) using CyberKnife robotic stereotactic radiotherapy. The median age of patients was 59 years (range 21–79 years) at diagnosis and 62 years (range 24–80 years) at radiotherapy. Most patients (57%) belonged to the age group between 51 and 69 years. Baseline ECOG grade was 0 or 1 in 160 patients and 2 in 12 patients. Seventy-two patients suffered from various neurological deficits, such as hemiparesis (18 patients) and visual impairment (36 patients). The most common location (55.2%) was the skull base, including cavernous sinus involvement (Table 1). 

All cases were discussed during a multidisciplinary board meeting. Patients with lesions less than 5 cm in largest diameter, with a well-defined border, no edema, and no signs of brain infiltration, qualified for radiotherapy with dose de-escalation.

In order to provide more transparency in statistical analysis, two groups were created: 67 patients (20 men and 47 women) in Group A had already undergone surgery as initial treatment before they were qualified for radiotherapy, whereas 105 patients (22 men and 83 women) in Group B were treated by robotic radiotherapy alone.

In Group A, gross total resection (Simpson I–III) was performed in 32 patients. The most common histological subtype was meningothelial meningioma WHO 1, especially in the cavernous sinus region (66%). Most atypical meningiomas were located on cerebral convexity. Table 2 shows the distribution of histological subtypes of tumors. Radiotherapy was implemented at the time of tumor recurrence. Thirty-five patients were irradiated due to subtotal tumor resection or biopsy (Simpson IV–V). The median time from surgery to the start of radiotherapy was 11 months (range 5–56 months) after STR (subtotal resection) and 59.5 months (range 14–211 months) after GTR, and the difference was statistically significant (*p* = 0.00000). 

Group B consisted of 105 patients treated by robotic hfSRT (hypofractionated stereotactic radiation therapy) without initial surgery. The diagnosis of meningioma and the qualification for radiotherapy is based on the tumor’s MRI features. The reason for radiotherapy in 41 patients was tumor progression on MRI scans (median time to progression 32 months, range 4–148 months). Sixty-four patients were qualified for radiotherapy shortly after the initial diagnosis because of tumor location (e.g., proximity of eloquent brain structures).

### 2.2. Radiotherapy Treatment Planning and Delivery

For all patients, a thermoplastic mask dedicated to stereotactic skull radiotherapy was prepared. Following that, thin-slice CT (computed tomography) and MRI with contrast enhancement were performed. Thirty-nine patients with skull base involvement had 68Ga-DOTATATE PET-CT (positron emission tomography-computed tomography) for better tumor visualization. All treatment series were registered using rigid algorithms. Radiotherapy plans were created using CyberKnife treatment planning software for both image registration and treatment planning (Multiplan 4.6 Accuray, Sunnyvale, CA, USA).

Gross Tumor Volume (GTV) was identical with tumor visible on MRI scans. Patients with brain infiltration did not qualify for the current study, so Clinical Target Volume (CTV) was identical with GTV. Due to the high precision of robotic radiotherapy no additional margin was added, and thus Planning Target Volume (PTV) was identical with GTV and CTV. PTV ranged between 0.8 and 29.3 cm^3^ (median 6.85 cm^3^).

Organs at risk (OAR) were contoured using usually T1 MRI sequence with contrast enhancement, which optimally defines the extension of meningioma (Figure 1). In some cases, other sequences were used for specific tumor location (e.g., T2 with fat saturation images for tumors with orbital involvement). OAR dose constraints were determined according to Timmerman et al.’s criteria as follows (point defined as a volume smaller than 0.035 cm^3^): Optic pathway (optic nerves, chiasm, and tracts): maximal point dose 17.4 Gy; 15.3 Gy to volume less than 0.2 cm^3^. Cochlea: maximal point dose 14.4 Gy. Brain stem: maximal point dose 23.1 Gy; 15.9 Gy to volume less than 0.35 cm^3^. Spinal cord: maximal point dose 22.5 Gy; 15.9 Gy to volume less than 0.35 cm^3^ [7].

As mentioned before, the total dose was 18 Gy delivered in three fractions. The dose was prescribed to the 78–92% isodose line. The minimal PTV dose was 11.8–17.8 Gy (median 17.2 Gy), the maximal PTV dose was 18.82–23.08 (median 20.69 Gy), and the mean dose was 17.05–20.71 Gy. The beam number ranged from 68 to 327 (median 168 beams). Patient positioning and treatment delivery was performed using CyberKnife System (Accuray, Sunnyvale, CA, USA). All patients completed radiotherapy in accordance with the treatment plan.

### 2.3. Follow-Up

MRI scans and clinical evaluations were obtained at 3, 6, and 12 months after radiotherapy, then yearly and after reaching five years follow-up period every other year. If any new neurological symptoms had appeared, or in the case of any diagnostic uncertainties, additional scans were performed. Twenty-two of the patients with meningiomas located in close proximity to visual pathways had additional detailed ophthalmological examination and follow-up.

### 2.4. Statistical Analysis

Overall survival (OS) and progression-free survival (PFS) were measured from the time of radiotherapy completion and the analysis was performed using the Kaplan–Meier method. The possible effect of several variables on PFS was determined using Cox regression (*p* values < 0.05 were considered significant). All statistical analysis was performed using Statistica 13.3 software.

## 3. Results

### 3.1. Local Control

Follow-up ranged from 18 to 124 months (median 67.5 months). Local control was achieved in 90.7% of patients—the tumor remained stable in 151 patients (87.8%), whereas five had partial regression (2.9%). Overall survival was 99.3% at 5 years and 98.5% at 8 years (Figure 2A). Five- and eight-year local control rates were 90.3% and 89.4%, respectively (Figure 2B).

We analyzed several factors that could potentially influence the outcomes. Five-year PFS was almost equal in all age groups: 24–50 years—91.2%, 51–69 years—90.2%, and 70–80 years—89.2%. Five-year local control rates were higher in women (91.5%) than in men (86.8%), but the difference was not statistically significant.

The worst outcome was observed for cerebral convexity meningiomas—5-year PFS at 75.5%—though it is worth noticing that 25% of tumors in this location were histopathologically confirmed atypical meningiomas. Five-year local control rates for other locations were as follows: 80% for optic nerve sheath meningiomas, 91.2% for skull base meningiomas, 92.8% for cavernous sinus meningiomas, 96.2% for falx meningiomas, and 100% for other locations. Tumor volume had no significant influence on PFS. 

Five-year PFS was significantly (*p* = 0.01) higher in Group B (radiotherapy alone) than in Group A (radiotherapy after initial surgery) and the rates were 95.9% and 81.8%, respectively (Figure 3).

In Group A we observed that local control rates at 5 years were higher for benign meningiomas (86.8%) than for atypical meningiomas (66.3%) (*p* = 0.03) (Figure 4). There was no significant difference in PFS between the histological subtypes. Five-year progression-free survival was 100% for angiomatous, transitional, psammomatous, and clear cell meningioma, 82.9% for meningothelial meningioma, 80% for fibrous meningioma, and 63.8% for atypical meningioma (*p* = 0.2). It must be stressed, however, that one of the less common subtypes was found in 11 patients from group A (angiomatous, transitional, psammomatous, or clear cell meningioma). When we only analyzed the three dominant subtypes (meningothelial, fibrous, and atypical), *p* was 0.07.

### 3.2. Patients with Tumor Progression

Local failure occurred in 16 patients (9.3%)—10 women and 6 men. The median time to progression was 27.5 months (range 12–62 months). Eleven patients underwent prior surgery (Group A)—four had GTR (and two of them had atypical meningioma) and seven had STR (in three of them meningioma was atypical). Five patients had no prior surgery (Group B) and the tumor progressed before radiotherapy.

After meningioma progression had been diagnosed, four patients were qualified for surgery. Three have stable disease (follow-up range 31–58 months), one patient showed further progression and eventually died 18 months after surgery.

Twelve patients underwent a second course of radiotherapy and were again given 18 Gy in three fractions. Median follow-up in this cohort was 29 months (range 6–54 months). Eleven patients had no progression, and one showed slow progression but died of heart failure 47 months after radiotherapy.

### 3.3. Tolerance of Treatment

Radiotherapy was very well-tolerated. Eighteen patients (10.5%) reported transient headaches, out of which 10 did not require any treatment (EORTC/RTOG scale grade 1). Only eight patients (4.7%) required small doses of glicocorticosteroids for a short period of time (EORTC/RTOG grade 2); in all those patients tumor volume was greater than 5 cm^3^ and the meningioma was localized on the cerebral convexity.

During the treatment planning process special attention was given to the dose to optic apparatus, as many tumors were localized either in the cavernous sinus or the optic nerve sheath. The median dose to the ipsilateral optic nerve was 15.62 Gy (range 3.10–17.33 Gy). In this group, an ophthalmological examination was performed on a regular basis, and 22 patients had extended evaluation, including the assessment of anatomical features (condition of eye surface, central corneal thickness, endothelial cell density, lens densitometry, central macular thickness, and retinal nerve fiber layer) and functional tests (visual acuity, intraocular pressure, visual field, and visual-evoked potentials) [8].

### 3.4. Clinical Effects of Radiotherapy

As shown in Table 3, 72 patients presented with neurological deficits before radiotherapy. During the follow-up, clinical symptoms were reassessed. After radiotherapy, a substantial improvement was observed in 27 patients (37.5% of those with previous neurological deficits), of which 23 patients reported complete regression of symptoms. Visual impairment had the biggest rate of symptomatic improvement (9 out of 36 patients reported complete regression of symptoms). Forty-three patients (59.7%) reported no change in pre-existing symptoms. Tumor progression only led to the deterioration of their neurological status for two patients (2.8%).

We also evaluated the influence of neurological status on treatment outcome and the results were statistically significant—5-year PFS for EORTC/MRC grade 1 (no symptoms), 2 and 3 were 96.8%, 81.9%, and 75.1%, respectively (*p* = 0.00153) (Figure 5).

### 3.5. Other Neoplasms Diagnosed during Follow-Up Period

During the follow-up period, 13 patients were diagnosed with other neoplasms: nine with breast cancer, two with lung cancer, one with colorectal cancer, and one with urinary bladder cancer. All cases of breast cancer were diagnosed as luminal A, with a strong expression of estrogen and progesterone receptors; they all received hormonal therapy, and no meningioma progression was observed in this group.

### 3.6. Univariate and Multivariate Analysis

We analyzed the influence of demographic, clinical, and histopathological factors on PFS: primary treatment (surgery vs. radiotherapy), age at diagnosis (<50 vs. >50 years), sex (female vs. male), gross tumor volume (<5 cm^3^ vs. >5 cm^3^), presence of neurological symptoms (no vs. yes), as well as extent of resection (Simpson I–III vs. IV–V) and histopathological grading (WHO 1 vs. WHO 2) in Group A, and tumor progression before radiotherapy in Group B (yes vs. no).

The univariate Cox regression identified that prior surgery, the presence of neurological deficits and (only in Group A) histopathological grading were associated with poorer outcome. The influence of age, sex, tumor volume, Simpson grade, and progression before radiotherapy was not statistically significant (Table 4). The multivariate Cox regression confirmed that the presence of neurological symptoms was an independent predictive factor of increased recurrence risk (HR 5.54) (Table 5).

## 4. Discussion

Meningiomas are the most common non-glial primary intracranial tumors. Most of them (80–90%) are benign, so the key is to choose a treatment which would lead to the achievement of the best local control rate and keep the morbidity at a low level. There is no general consensus for the total dose and fractionation in radiotherapy of meningiomas. We decided to evaluate the effects of hypofractionated stereotactic radiation therapy with the lowest commonly acceptable dose of 18 Gy in three fractions.

### 4.1. Study Population

In our study, the majority of patients were women (75.6%), and the female to male ratio was 3.1:1. The same proportion was reported in most studies [2,9,10,11,12]. Similar to other analyzed populations, most meningiomas were diagnosed in the sixth or seventh decade of life [2,9,10,11]. In the literature, atypical meningiomas are reported to be more common in the male population [13]. We also found in Group A that meningiomas WHO 2 accounted for 30% cases in men, but only for 21.3% cases in women.

Park et al. observed the meningothelial subtype to be most common, followed by transitional and fibrous [14]. Almost half of the tumors in our study, after prior surgery (Group A), were meningothelial, followed by fibrous and transitional. When analyzing the relationship between location and histopathological grading, most atypical tumors were located on cerebral convexity and skull base, just as Bhat et al. reported [15].

One patient had the diagnosis of neurofibromatosis 2, and was the youngest of all the patients in our group (21 years at diagnosis); she underwent prior surgery and had a fibrous subtype of the tumor. Meningioma in patients with neurofibromatosis 2 is known to usually be diagnosed at a younger age than in the general population and the histopathological subtype is mostly fibrous or transitional [2,16]. Although some researchers [17] report a more aggressive nature of meningioma in patients with NF2 mutation, in this case we did not observe tumor progression.

In some studies, the coincidence of meningiomas and other neoplasms is described: foremost breast cancer, but also lung and urinary bladder cancer. It has to be stressed, however, that there is no cause-and-effect relationship, but rather those tumors share some risk factors [1,18,19,20,21]. Nine of our female patients developed breast cancer and, due to type luminal A, all of them were given hormonal therapy. No meningioma progression was seen in this group; this could support the results of some studies describing the effects of antiestrogenic therapy of meningiomas [22,23,24,25,26].

### 4.2. Treatment Results

Meningiomas, especially those with the largest diameter smaller than 3 cm which were inoperable due to their location or the patient’s performance status, are the ideal target for radiosurgery, whose results, according to some studies, are comparable with radical surgery (Simpson I) [27,28]. Larger tumors or those in proximity to organs at risk (e.g., optic nerves and chiasm) can be treated with hypofractionated stereotactic radiotherapy, which combines the advantages of other treatment modalities: steep dose falloff and short treatment time such as in radiosurgery, and the possibility of repair between fractions and decreased risk of complications, as is typical for conventional radiotherapy [3,29,30,31].

In our study, all patients underwent hfSRT with CyberKnife and received 18 Gy in three fractions. The location of most meningiomas was the skull base (55.2%). Tumor volume ranged from 0.8 cm^3^ to 29.3 cm^3^ (median 6.85 cm^3^). During the follow-up, local control was achieved in 90.7% of patients. Oh et al. presented the results of large meningiomas treatment in 31 patients. All were greater than 10 cm^3^ (range 11.6–58.2 cm^3^) and were treated with a dose of 22.6–27.8 Gy in 3–5 fractions. Partial regression was seen in 54.8% [32]. Tuniz et al. treated a variety of tumors (meningiomas, glomerulomas, neuromas) with a volume greater than 15 cm^3^. After receiving 18–25 Gy in 2–5 fractions, no patient showed tumor progression during the median follow-up of 31 months (range 12–77 months) [33]. Similarly, local control in 100% of patients was achieved during the median follow-up of 60 months by Conti et al. by using a similar fractionation schedule (18–25 Gy in 2–5 fractions), although the tumors were significantly smaller (median volume 4.95 cm^3^) [34]. In 96 patients, Meniai-Merzouki et al. also used various fractionationschemese. The median dose was 25 Gy in five fractions (range 16–40 Gy in 3–10 fractions). Five-year local control was 74%, and in patients who received doses higher than the median dose, 5-year PFS was 88% (median follow-up time was 20.3 months) [35]. Table 6 shows the comparison of treatment outcome achieved in our material and the studies mentioned above, taking into account the biological effective dose (BED). For the analysis we used the alpha/beta ratio 3.28 and 3.76, calculated by Shrieve and Vernimmen, respectively [36,37].

Seventy-two of our patients presented with neurological deficits and after radiotherapy completion 37.5% experienced symptomatic improvement or complete resolution of symptoms. The results were similar to those reported by Meniai-Merzouki et al. [35], yet our patients reported more positive effects than in the study by Conti et al. [34], despite a similar BED. Oh et al. observed improvement in 95.2% of patients [32], however BED was 47% higher than in our study. Table 7 shows the comparison of clinical effects in our study and in other abovementioned studies.

### 4.3. Treatment Tolerance

Only 10.5% of our patients reported headaches and less than half of them (4.7%) required small doses of corticosteroids for a short period of time (EORTC/RTOG grade 2). We did not observe any late complications. Santacroce et al., analyzing a group of more than four thousand patients treated with GammaKnife radiotherapy (median dose 14 Gy), described early effects in 12.8% of patients and late effects in 4.8% [38]. Similar observations were made by Przybylowski et al.—early effects occurred in 8.3% of patients after a median radiation dose of 15 Gy [39]. Harat et al. treated meningiomas, arteriovenous malformations, and cerebral metastases with a median dose of 16 Gy and described cerebral edema, which lasted up to 6 months, in 17% of patients [40]. It is worth noting that the edema occurred mostly around tumors located above the Frankfurt line; this matches our experience, as patients who required corticosteroids had tumors located on brain convexity. Using a slightly higher radiation dose (21–23 Gy) compared to our study, Meniai-Merzouki et al. observed early effects in 34% and late effects in 2% of patients [35].

### 4.4. Prognostic Factors

Simpson et al., Narayan et al., Champeaux et al., and other authors view radical neurosurgery as the most important prognostic factor [5,41,42,43,44]. In Group A of our study, before the onset of radiotherapy, the difference in progression-free survival between the subgroup after GTR and the subgroup after STR was 48.5 months and was statistically significant (*p* = 0.00000).

Prior to surgery, according to many authors, worse treatment outcomes were forecast [40,44,45,46]. We observed a significant difference in five-year PFS (*p* = 0.016) between Group A—81.8% and Group B—95.9%.

Age at diagnosis below 50 years seems to be a significant prognostic factor, according to Champeaux et al., Fokas et al., and Zaher et al. [45,47,48], but the differences mainly apply to overall survival. In our study, the differences of PFS rates between particular age groups were smaller than 2%, but due to low mortality (1.2%) we did not compare overall survival in those groups.

In our study, we observed slightly lower PFS rates in the male population. Although the difference was not significant, Solda et al., Santacroce et al., dos Santos et al., and Zhang et al. described male sex as negative prognostic factor [38,44,49,50].

Santacroce et al. and Wang et al. indicated histopathological WHO grade as a prognostic factor in the treatment of meningiomas. We also stated a statistically significant difference in PFS for WHO 1 and WHO 2 meningiomas—86.8% and 66.3%, respectively [38,51].

Five-year PFS for meningiomas greater and smaller than 5 cm^3^ was assessed and the difference was 3% and not significant. Kondziolka et al. described a worse outcome for tumor volume greater than 7.5 cm^3^, and a similar outcome was observed by Pollock et al. and Zhang et al. [44,52,53].

The presence of neurological symptoms turned out to be an independent prognostic factor, increasing the risk of progression by more than fivefold (*p* = 0.0095). A similar relation was observed by Kondziolka et al. and Kępka et al. [53,54]. Zhang et al. and Soyuera et al. indicated that the performance status of a patient is an important prognostic factor [44,55].

## 5. Conclusions

Based on the results of our study, robotic stereotactic radiotherapy with a total dose of 18 Gy delivered in three fractions has a similar efficacy to radiotherapy schedules with higher BED, which can support the application of dose de-escalation in the treatment of intracranial meningiomas. By giving a lower radiation dose, it is also possible to administer a second course of radiotherapy in the case of progression. Crude progression free survival after both primary and secondary treatment was 98.8%. The treatment is well-tolerated, with a low risk of early and late effects. The presence of neurological symptoms before the onset of treatment is an independent prognostic factor, increasing the risk of progression.

## Figures and Tables

**Figure 1 cancers-15-05436-f001:**
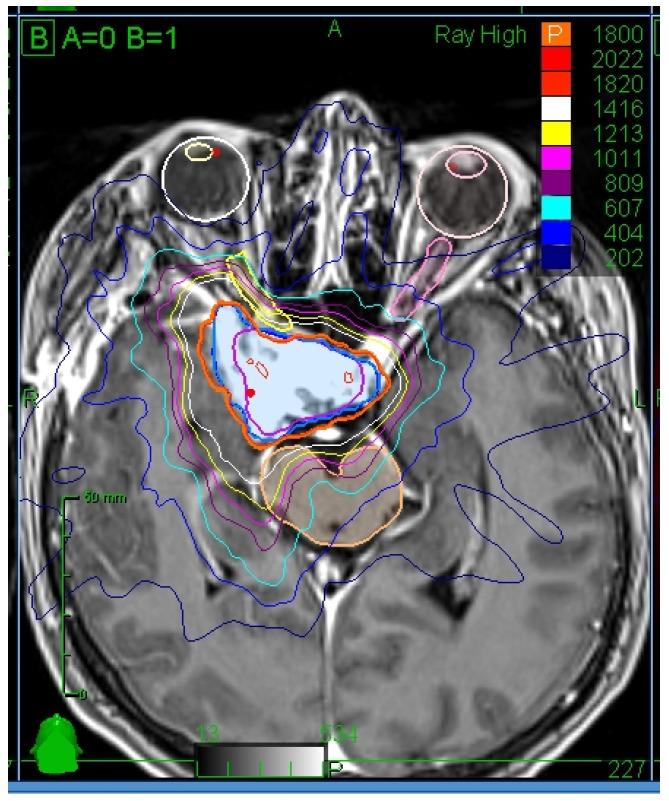
Illustration of GTV and OAR contours and isodoses. Axial view.

**Figure 2 cancers-15-05436-f002:**
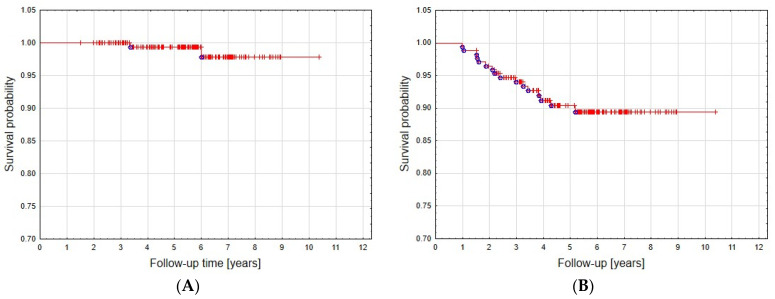
Overall survival (**A**) and local control (**B**) curve.

**Figure 3 cancers-15-05436-f003:**
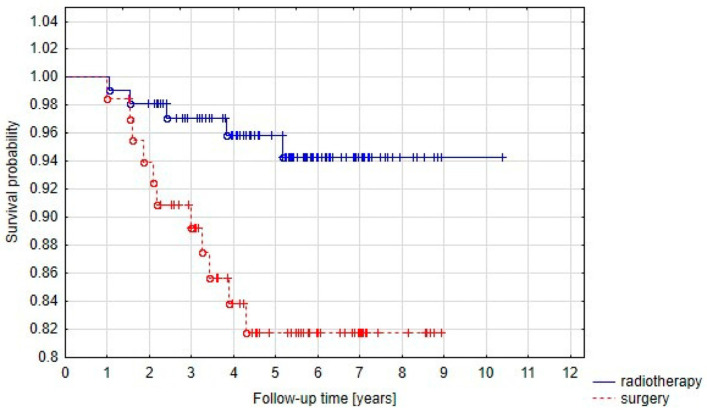
Progression-free survival curve in Group A (prior surgery) and Group B (radiotherapy alone).

**Figure 4 cancers-15-05436-f004:**
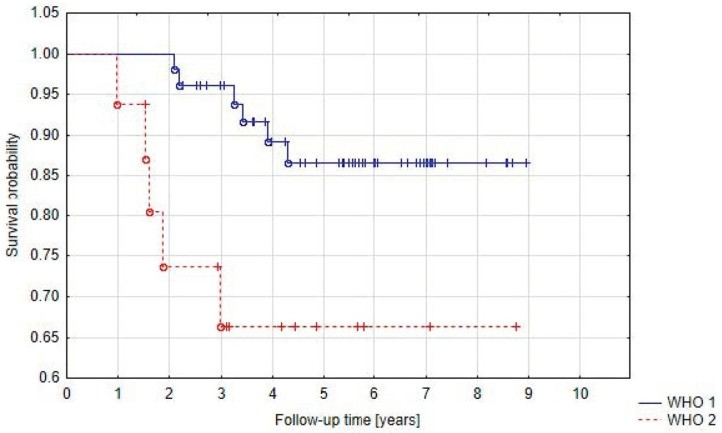
Progression-free survival in Group A related to WHO grade.

**Figure 5 cancers-15-05436-f005:**
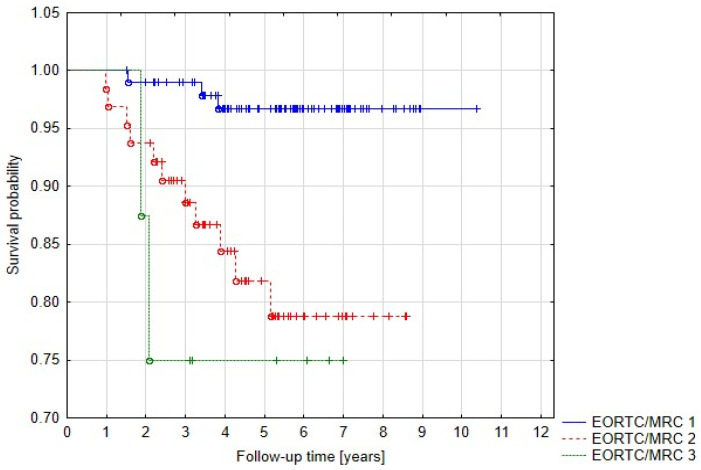
Progression-free survival related to neurological status.

**Table 1 cancers-15-05436-t001:** Topography of meningiomas.

Topography	Number of Patients	%
Skull base	95	55.2
(incl. cavernous sinus)	(42)	(24.4)
Falx or parasagittal	30	17.4
Convexity	28	16.3
Cerebellopontine angle	8	4.7
Optic nerve sheath	6	3.5
Lateral fissure	5	2.9

**Table 2 cancers-15-05436-t002:** Distribution of histological subtypes in Group A.

	Subtype	Number of Patients	%
WHO 1	Meningothelial	32	47.8
	Fibrous	9	13.4
	Angiomatous	2	3
	Psammomatous	2	3
	Transitional	6	8.9
WHO 2	Atypical	15	22.4
	Clear Cell	1	1.5

**Table 3 cancers-15-05436-t003:** Neurological symptoms before radiotherapy and during follow-up.

Symptom	Number of Patients	Complete Regression	Improvement	Worsening
Hemiparesis	18	6	1	2
Visual impairment	36	9	-	-
Hearing impairment	3	-	-	-
Facial nerve palsy	12	7	1	-
Epilepsy	2	-	2	-
Severe headaches	1	1	-	-

**Table 4 cancers-15-05436-t004:** Univariate Cox regression (PFS).

Prognostic Factor	5-Year PFS	*p*	HR	CI (95%)
Prior surgery:		0.016		
-yes	82%		3.67	1.27–10.55
-no	96%			
Age at diagnosis:				
<50 years	86%	0.335	0.61	0.22–1.67
>50 years	91%			
Sex:				
-female	92%	0.19	0.51	0.18–1.4
-male	87%			
GTV:				
<5 cm^3^	92%	0.298	1.82	0.59–5.66
>5 cm^3^	89%			
Presence of neurological symptoms:				
-no	96%	0.00226	7.09	2.02–24.95
-yes	81%			
Group A				
Simpson Grade:				
I-III	86%	0.34	1.82	0.53–6.22
IV-V	77%			
Histology:				
WHO 1	87%	0.027	3.86	1.17–12.72
WHO 2	66%			
Group B				
Progression prior to radiotherapy:				
-yes	89%	0.089	6.66	0.74–59.63
-no	100%			

**Table 5 cancers-15-05436-t005:** Multivariate Cox regression (PFS).

Prognostic Factor	*p*	HR	CI (95%)
Surgery and radiotherapy vs. radiotherapy alone	0.13	2.32	0.78–6.88
No symptoms vs. neurological deficits	0.0095	5.54	1.52–20.23

**Table 6 cancers-15-05436-t006:** Comparison of different schedules of hypofractionated robotic stereotactic radiotherapy, with regard to BED.

Study	Median Total Dose [Gy]	Number of Fractions	Median Follow-Up [Months]	Local Control [%]	BED Alpha/Beta = 3.28	BED Alpha-Beta = 3.76
Grzbiela et al.	18	3	67.5	90.7	50.9	46.7
Oh et al. [32]	27.8	5	57	90.3	74.9	68.9
Tuniz et al. [33]	24	3	31	100	82.5	75.1
Conti et al. [34]	23	5	60	100	55.3	51.1
Meniai-Merzouki et al. [35]	25	5	20.3	74	63.1	58.2

**Table 7 cancers-15-05436-t007:** Clinical effect of hypofractionated robotic stereotactic radiotherapy.

Study	Complete Regression [%]	Symptomatic Improvement [%]	Worsening [%]
Grzbiela et al.	31.9	5.6	2.8
Oh et al. [32]	95.2		Not available
Tuniz et al. [33]	21		0
Conti et al. [34]	0	18	0
Meniai-Merzouki et al. [35]	37	14	5

## Data Availability

The data are not publicly available due to confidentiality and ethical considerations.

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
