# Peer review of "Robotic Stereotactic Radiotherapy for Intracranial Meningiomas—An Opportunity for Radiation Dose De-Escalation"

_cancers, 2023, doi:10.3390/cancers15225436_

Round 1
Reviewer 1 Report
Comments and Suggestions for Authors
In this study, robotic stereotactic radiotherapy with total dose of 18 Gy delivered in three fractions has similar efficacy to conventional radiotherapy schedules in meningiomas. The treatment is well tolerated, with low rate of adverse events.
However, the evaluation of local tumor control in a tumor that has a low malignant potential is of moderate interest, as many of these tumors will show only a slow to no progression. Neurologic symptoms can be present in a percentage of patients, and these are the ones that can benefit from the novel treatment approach.
How were patientes allocated to either Group A or B? What were the criteria for surgical resection? Are histologic reports only availabel for patients that underwent surgery? As mentioned, several subtypes of meningioma exist. Do all subtypes equally respond to radiation therapy? Allocation of patient subgroups should be explained more in detail, to make clear that no selection bias is present.
Author Response
Thank you very much for taking the time to review the manuscript. Please find the detailed responses below and the corresponding corrections highlighted in the re-submitted file.
Comments 1: However, the evaluation of local tumor control in a tumor that has a low malignant potential is of moderate interest, as many of these tumors will show only a slow to no progression. Neurologic symptoms can be present in a percentage of patients, and these are the ones that can benefit from the novel treatment approach.
Response 1: Meningiomas are mostly benign tumors, which show a high incidence, yet at the same time the majority is asymptomatic and is found accidentally during MRI performed for other reasons. However, some of them may progress and cause neurological symptoms corresponding with the tumor location. In our material, in Group A, gross total resection was performed in 32 patients and after a median time of 59.5 months (range 14-211 months) tumor recurrence was diagnosed (paragraph 2.1, page 3, line 8-12). In Group B, 41 patients were irradiated due to tumor progression (paragraph 2.1, page 3, line 19-21). Therefore, in 73 patients (42.4%) radiotherapy was implemented to a progressive disease. Patients who had no tumor progression received radiotherapy to prevent either the onset or worsening of neurological symptoms.
Comments 2: How were patients allocated to either Group A or B? What were the criteria for surgical resection? Allocation of patient subgroups should be explained more in detail, to make clear that no selection bias is present.
Response 2: Groups A and B were created solely to provide more transparency in statistical analysis, as only in Group A some prognostic factors, like tumor’s histology, could have been evaluated. The patients were not randomized or allocated to those groups during the multidisciplinary board meetings. At the time they were qualified for radiotherapy, the surgery had been already performed (5-211 months before the start of radiotherapy). Thank you for this comment, the corresponding section has been rephrased to make it more clear: “In order to provide more transparency in statistical analysis, two groups were created: 67 patients (20 men and 47 women) in Group A had already underwent surgery as initial treatment before they were qualified for radiotherapy, whereas 105 patients (22 men and 83 women) in Group B were treated by robotic radiotherapy alone.” (paragraph 2.1, page 3, line 4-7).
Comments 3: Are histologic reports only availabel for patients that underwent surgery?
Response 3: Yes, histological reports are only available for patients that underwent surgery. In meningiomas, biopsy is usually not necessary as a part of the diagnostic process, because in most cases the features of meningioma on MRI are sufficient for diagnosis. In Group A, there was one patient who underwent biopsy (“Thirty-five patients were irradiated due to subtotal tumor resection or biopsy (Simpson IV–V).”, paragraph 2.1, page 3, line 12-13), however, biopsy was the result of a surgery that was originally planned as subtotal tumor resection. In Group B, there were no histological reports, as “the diagnosis of meningioma and the qualification for radiotherapy based on tumor’s MRI features” (paragraph 2.1, page 3, line 18-19).
Comments 4: As mentioned, several subtypes of meningioma exist. Do all subtypes equally respond to radiation therapy?
Response 4: We found no significant difference in progression-free survival between the histological subtypes, as stated in paragraph 3.1, page 6, line 2-3.
Five-year progression-free survival was 100% for angiomatous, transitional, psammomatous and clear cell meningioma, 82.9% for meningothelial meningioma, 80% for fibrous meningioma and 63.8% for atypical meningioma (p=0.2). It must be stressed, however, that only in 11 patients from group A one of the less common subtypes was found (angiomatous, transitional, psammomatous or clear cell meningioma). When we analyzed only the three dominant subtypes (meningothelial, fibrous and atypical) p was 0.07.
Considering your question, we decided to enclose these detailed results in our article (paragraph 3.1, page 6, line 3-9).
Reviewer 2 Report
Comments and Suggestions for Authors
Review
Robotic Stereotactic Radiotherapy for Intracranial Meningiomas — the Possibility of Radiation Dose Reduction
Titel perhaps better / nicer: “an opportunity for radiation dose deescalation?”
All together:
Very interesting data, well described collective.
Interesting dose prescribtion, especially in comparison with the other cited SRS series, but also in comparison with conventional radiotherapy (e.g. 25 x 2 Gy, BED about 100 Gy, while 3 x 6Gy would result in a BED of round about 72 Gy).
I have only one minor remark:
Abstract
“Four patients were operated due to disease progression…”
please rephrase: “4 patients required surgery due to progressive disease….”
Author Response
Thank you very much for taking the time to review our manuscript and for your suggestions. Please find the detailed responses below and the corresponding corrections highlighted in the re-submitted file.
Comments 1: Robotic Stereotactic Radiotherapy for Intracranial Meningiomas — the Possibility of Radiation Dose Reduction. Titel perhaps better / nicer: “an opportunity for radiation dose deescalation?”
Response 1: Thank you for your recommendation. The title has been rephrased in accordance with your suggestion: “Robotic Stereotactic Radiotherapy for Intracranial Meningiomas — an Opportunity for Radiation Dose De-escalation.”
Comments 2: Abstract: “Four patients were operated due to disease progression…” please rephrase: “4 patients required surgery due to progressive disease….”
Response 2: Thank you for this remark. The sentence has been rephrased as you suggested (in Abstract, section Results): “Four patients required surgery due to progressive disease, three of them are progression free during further follow-up.”